# The Aggressive Gender Backlash in Intimate Partner Relationships: A Theoretical Framework and Initial Measurement

**DOI:** 10.3390/bs15070941

**Published:** 2025-07-11

**Authors:** Aristides A. Vara-Horna, Noelia Rodríguez-Espartal

**Affiliations:** 1Faculty of Administrative Sciences and Human Resources, San Martín de Porres University, Lima 15009, Peru; 2Faculty of Psychology, Pontifical Catholic University of Peru, Lima 15088, Peru; noelia.rodrigueze@pucp.edu.pe

**Keywords:** aggressive gender backlash, intimate partner relationships, empowerment, emotional coercion, violence prevention, psychometric validation, General Aggression Model, women’s autonomy

## Abstract

This study introduces and validates a novel instrument to measure aggressive gender backlash (AGB), a distinct and underexplored dimension of gender backlash (GB) within intimate partner relationships. Based on the General Aggression Model, a multidimensional scale was developed and tested using data from 513 Peruvian female microentrepreneurs. Results demonstrate solid evidence of reliability, discriminant validity, and predictive validity across five dimensions: hostility, the withdrawal of support, sabotage/coercion, gender stereotyping, and masculine victimization. The findings reveal that AGB is more prevalent than intimate partner violence against women (IPVAW) and often precedes it. AGB encompasses covert, non-violent behaviors that aim to resist female empowerment, such as emotional sabotage, manipulation, and disqualification, often normalized within relationships. This construct is significantly associated with lower levels of empowerment, increased subordination, emotional morbidity, and decreased work productivity. This study redefines GB as an interpersonal process measurable at the individual level and provides the first validated tool for its assessment. By conceptualizing AGB as a persistent, harmful, and functionally equivalent mechanism to IPVAW, though not necessarily physically violent, this research fills a key gap in gender violence literature. It offers practical implications for early detection and prevention strategies.

## 1. Introduction

Intimate partner violence against women (IPVAW), a phenomenon of great magnitude and severity at the global level, has been the focus of numerous investigations in recent decades. These studies have aimed to identify the causes and consequences of this phenomenon and develop effective strategies for its prevention and eradication ([28]; [39]). Within the broad spectrum of this issue, the intersection between female empowerment and violence represents one of the most complex challenges for research. Although theory suggests that female empowerment should decrease violence by partners or former partners, empirical findings often demonstrate a more complicated and, in some instances, counterintuitive relationship, with violence persisting or even intensifying ([14]).

In this context, the concept of “gender backlash” (GB) emerges as a valuable theoretical tool. GB posits that men’s resistance to changes in power relations can manifest as violence and hostility towards women, particularly in contexts where women challenge traditional gender norms ([2]; [9]; [13]; [17]; [19]; [20]; [27]; [42]; [44]; [48]). However, despite its analytical power, GB remains underdeveloped as a measurable construct.

The term “backlash” was first introduced in political science in the 1970s through the work of Lipset and Raab, and later elaborated by Mansbridge. Initially, it was understood as a reactionary countermovement resisting the advancement of rights ([2]). In sociological terms, backlash refers to a series of practices and processes that aim to perpetuate inequalities by reacting defensively to perceived threats against privileged social statuses.

In the gender domain, [17] ([17]) introduced the term “gender backlash” to describe male resistance aimed at preserving patriarchal power structures amidst the progress of feminist movements in the United States. [27] ([27]) further defined it as the use of coercive power to regain lost status. While they did not detail the mechanics of this reaction, they emphasized its potential range, from subtle ridicule and condemnation to overt physical violence. Their framework distinguished between violent and hostile responses, contributing significantly to the concept’s theoretical boundaries.

Contemporary research aligns with this perspective, conceptualizing GB as an adverse reaction to women’s empowerment driven by the desire to restore male dominance, often rooted in ideology and amplified by emotional responses such as fear, anger, and resentment ([2]; [17]; [19]; [10]; [27]; [45]; [46]).

Despite its growing theoretical appeal, GB has been employed in empirical research primarily as a proxy variable—indirectly captured through gender disparities in income or education, or hostile reactions to the violation of gender norms ([6]; [8]; [18]; [22]; [23]; [26]; [29]; [32]; [35], [36]; [34]; [37]; [49]). These approaches, however, suffer from significant limitations in terms of construct validity and reliability, as they do not capture the nuanced interpersonal dynamics through which GB is enacted.

This study addresses this gap by proposing and validating a pioneering instrument to measure a specific dimension of GB: aggressive gender backlash (AGB). Unlike existing proxy measures, this scale captures covert, intentional, and non-violent forms of resistance to female empowerment within intimate partner relationships. By conceptualizing AGB as an interpersonal process measurable at the individual level, this study enhances our understanding of how resistance to gender equality materializes in everyday relational contexts and offers a tool for the early detection and prevention of gender-based harm.

### 1.1. Aggressive Gender Backlash: Theoretical Proposal

Despite its growing relevance, gender backlash (GB) remains theoretically underdeveloped. A critical review of the literature reveals three core insights: (1) GB is a reactive form of resistance to women’s empowerment, grounded in perceived threats to male privilege; (2) it spans a continuum of behaviors, often covert and socially normalized, thus escaping conventional violence classifications; and (3) it is affectively and ideologically driven, rooted in emotions like anger or fear and reinforced by gendered beliefs about power and entitlement.

Building on these premises, this study introduces a conceptual distinction between violent GB—which includes physical, sexual, or psychological violence—and aggressive gender backlash (AGB), which refers to non-violent but intentional behaviors such as emotional withdrawal, subtle sabotage, blame, or manipulation. While AGB does not meet legal thresholds for violence, it serves similar functions: resisting empowerment and restoring hierarchical gender dynamics.

This refinement enables AGB to be analyzed as an independent construct, distinct from intimate partner violence against women (IPVAW). It shifts attention from overt violence to chronic, insidious resistance mechanisms that operate within seemingly non-violent relationships.

Accordingly, the framework posits nine theoretical premises, starting with proposition *1. Not all GB is violent.* As noted by [15] ([15], [16]), [38] ([38]), [40] ([40]), and [47] ([47]), GB is often invoked to explain IPVAW in the context of women’s empowerment. However, this study reconceptualizes GB as a multidimensional phenomenon that includes—but is not limited to—violence. Violent GB entails overt aggression aimed at controlling empowered women, whereas AGB involves covert, non-violent yet functionally aggressive acts ([1]). Differentiating the two avoids circular reasoning and allows AGB to be examined as a mechanism of relational subordination. This approach opens new avenues for understanding how power is preserved and resistance is enacted in intimate relationships without physical violence.

*2. AGB is different from psychological violence.* Although aggressive gender backlash (AGB) shares some characteristics with traditional psychological violence, such as emotional manipulation, there are crucial differences. While psychological violence intentionally seeks to cause deep emotional harm through direct insults, constant humiliation, or explicit intimidation, AGB is aimed explicitly at resisting and counteracting female empowerment through more subtle and socially acceptable strategies, such as withdrawing emotional support, indirectly sabotaging personal and professional goals, or blaming women for not fulfilling traditional roles (see Table 1).

*3. AGB is a precursor and risk factor for IPVAW.* Although less intense and rarely sanctioned, AGB shares core dynamics with intimate partner violence against women (IPVAW) and can act as a precursor. Through manipulation, emotional withdrawal, and subtle hostility, AGB functions as a low-risk but harmful strategy to reassert control in intimate relationships. These behaviors, often normalized, can escalate over time.

Several mechanisms link AGB to subsequent violence. First, it reinforces traditional gender roles, fostering conditions conducive to IPVAW. Second, repeated aggressive behaviors may intensify, evolving into more overt control or abuse. Third, AGB facilitates emotional harm that, while less visible, can profoundly affect mental health. Finally, prolonged exposure to AGB undermines women’s self-esteem and resilience, increasing vulnerability to future violence.

Thus, AGB not only mirrors some dynamics of IPVAW but may lay the groundwork for its emergence. Addressing AGB early is critical to breaking the cycle of escalating coercion and preventing more severe forms of gender-based violence.

*4. AGB is more prevalent than IPVAW.* AGB appears to be more prevalent than overt violence against women, largely due to its subtlety and normalization. Several sociocultural factors contribute to this pattern. First, many AGB behaviors—such as emotional manipulation or withdrawal—are socially tolerated and even perceived as acceptable in intimate relationships. Second, perpetrators often underestimate the harm caused by these actions, as they fall outside conventional definitions of violence. Third, the low likelihood of social or legal consequences reduces deterrence. Fourth, AGB tends to manifest in subtle ways that are difficult to detect or report. Finally, individuals may adopt AGB behaviors as a means of preserving their self-image, avoiding overt violence while still resisting gender equality. These factors collectively explain why AGB may occur more frequently and go largely unchallenged in everyday relational contexts.

*5. The General Aggression Model can be used to understand AGB.* The General Aggression Model (GAM) theory, proposed by [3] ([3]), parsimoniously integrates the various theories of human aggression, intending to explain how personal and situational factors influence aggressive behavior through cognitive appraisal, negative affect, and physiological activation. These same processes can be applied to AGB, involving emotional and cognitive activation states, cognitive appraisal processes, and hostile and instrumental gender responses (see Table 2).

The General Aggression Model (GAM) provides a comprehensive framework for explaining aggressive gender backlash (AGB), integrating cognitive, emotional, and situational processes. Unlike structural theories—such as Control Theory ([41]) or feminist analyses of patriarchy ([11])—GAM focuses on how these dynamics unfold at the individual and interpersonal levels through internal pathways.

GAM distinguishes between proximal and distal processes. The proximal processes—inputs, internal states, and outcomes—are especially relevant for understanding episodic aggression such as AGB ([1]):

*Inputs.* Women’s empowerment may trigger frustration in men, particularly those socialized with hegemonic masculinity, low self-esteem, or tolerant attitudes toward IPVAW. This perceived threat to male privilege acts as a catalyst for AGB.

*Internal pathways.* Frustration activates psychophysiological arousal and hostile emotions (e.g., anger and indignation), which are intensified by cognitive distortions like false attribution (“she’s to blame for my discomfort”) or false justice (“her autonomy is unfair to me”). These responses often reflect a loss of unacknowledged social privilege.

*Outcomes.* A cost–benefit appraisal follows. When aggressive behavior is seen as justifiable and low-risk—especially in the case of AGB, which is normalized and rarely sanctioned—it becomes more likely. These actions then reinforce the cycle, increasing the probability of recurrence.

According to [3] ([3]), two behavioral patterns are typical responses to threatened male dominance: 1. Systematic hostility. Passive yet persistent behaviors like emotional withdrawal, silence, or sabotage erode women’s autonomy. 2. Coercive gender manipulation. More strategic efforts to reassert control through blame, pressure, or appeals to traditional roles.

Both strategies reflect gendered efforts to regain power. If unsuccessful, they may escalate into more overt violence, though with higher social and legal costs. GAM thus offers a robust explanatory model for how non-violent but intentional gendered aggression, such as AGB, emerges and persists.

*6. AGB has a graduated behavioral intensity spectrum.* [19] ([19]) propose a spectral classification of GB. Although it is not aimed at explaining why AGB occurs, but rather at understanding how men may oppose prevention and gender activism, their spectrum of resistance ordered by intensity is very useful as a model for developing indicators of AGB. Using the analogy of [19] ([19]), AGB can be understood as a behavioral spectrum, in other words, as a set of behaviors that share the same motivation: regaining power. Still, the intensity can vary, and if a certain threshold is exceeded, it can lead to violence. As the spectrum becomes more intense, the probability of IPVAW increases. In the case of AGB, behaviors with lower intensity (more passive) may be showing annoyance, torturous silence, or reducing social support. Then, with medium intensity, there may be recrimination, minimization, stereotyping, manipulation, and blaming. Finally, with high intensity (more active), there may be sabotage, aggressive transference, aggressive treatment, and coercion.

*7. AGB is a second-order latent variable.* Whereas most empirical studies treat GB as an observable phenomenon—typically measured through resource discrepancies between partners ([4])—this study conceptualizes AGB as a second-order latent construct. Rather than a single behavior, AGB is inferred from multiple interrelated indicators rooted in psychological processes and expressed with varying intensities. This approach captures the complexity of AGB and allows for the examination of its underlying dimensions as part of a broader resistance to women’s empowerment.

AGB is a behavioral process enacted by male partners to resist female empowerment. While it can be perceived, its effects may also operate below the level of conscious recognition, particularly in contexts where such behaviors are normalized. It manifests itself through emotionally and cognitively charged hostile and instrumental gender behaviors. Emotionally, anger and a sense of loss of control or power can trigger this behavior. Cognitively, irrational ideas and gender biases form the basis for blaming women. These emotions and thoughts may manifest in behaviors with hostile or instrumental orientations.

At the hostile level, AGB acts as a form of impulsive aggression associated with anger and impulsivity, with the desire to dominate others through harm ([7]). This dimension can vary in intensity, ranging from showing annoyance and reprimanding to minimizing and aggressive treatment, ultimately manifesting as aggressive transference. In another dimension, AGB focused on gender control acts as a form of instrumental aggression, whose objective is not to inflict harm but to obtain benefits at the expense of the other person ([5]). This dimension can also vary in intensity from reducing support and sabotage to coercion.

At the cognitive level, AGB is often motivated by cognitive distortions. The loss of power is perceived as a threat to the privileged position, with attributions of guilt toward the partner (false attributions) or a strong sense of injustice (false justice). These irrational beliefs may intensify hostility and the rumination of anger ([12]). Women may report that their partners berate them for not fulfilling their gender roles or for threatening their masculinity and privilege, reflecting blaming stereotypes and perceptions of affecting their masculinity.

*8. AGB has a semi-circumplex structure.* This study proposes that Aggressive Gender Backlash (AGB) follows a semi-circumplex structure, organized along two underlying dimensions: intensity (passive to active) and dominant internal state (emotional, cognitive, or behavioral). This model captures the non-dichotomous nature of AGB, showing its gradual variation and interaction with broader relational factors such as power, empowerment, and violence.

AGB behaviors can be positioned as follows: On the behavioral side, AGB can manifest itself by withdrawing or decreasing support, interfering with the partner’s plans, or coercing the partner to abandon them. Each of these actions is positioned in the model according to its level of intensity. Emotionally, AGB can present itself in various ways, depending on the degree of passivity or activity. The most passive manifestation could be a non-explicit discomfort and rumination/irritation towards the partner’s behavior. At an intermediate level, the behavior becomes aggressive, more expressive, and directed towards the partner; at the most active level, the AGB may manifest itself through transferential aggression towards the partner’s belongings. Cognitively, AGB manifests itself differently according to its level of activity. In more passive states, cognitions focus on distancing from the partner’s activities and minimizing or devaluing them. At intermediate levels, cognitions blame the partner for neglecting the home or the relationship. At more active levels, cognitions focus on self-victimization, interpreted as direct damage to masculinity.

*9. AGB is as effective as IPVAW in subordinating women and limiting their empowerment, and as harmful due to its chronic nature*. While intimate partner violence against women (IPVAW) exerts control through overt, acute, and often legally punishable harm, aggressive gender backlash (AGB) operates through covert, repetitive, and socially normalized behaviors. These actions—such as emotional withdrawal, sabotage, persistent blame, and cognitive undermining—function as chronic stressors that accumulate over time. Rather than relying on episodic violence, AGB exerts continuous pressure that systematically erodes women’s empowerment, autonomy, emotional well-being, and productivity. In this sense, AGB constitutes a long-term mechanism of relational subordination with substantial psychosocial consequences, often as disabling as those resulting from physical or psychological violence ([44]).

This hypothesis challenges the implicit hierarchy of harm that places physical or psychological violence above non-violent forms of aggression. It proposes that the chronic and insidious character of AGB enables it to achieve comparable, and in some cases greater, effects in maintaining gender subordination. By conceptualizing AGB as both functionally and affectively equivalent to IPVAW, this proposition broadens current understandings of how intimate partner power dynamics are sustained and how damage is inflicted beyond conventional violence.

### 1.2. Research Objectives and Hypotheses

The present study aims to conceptualize and empirically validate aggressive gender backlash (AGB) as a distinct construct reflecting non-violent but intentional resistance to women’s empowerment within intimate partner relationships. Specifically, the aims of this research are as follows:Establish the psychometric properties of the AGB scale, including its reliability and its dimensional structure.Provide evidence of the AGB scale’s discriminant validity by distinguishing it from intimate partner violence against women (IPVAW).Assess the scale’s predictive validity by examining the relationship between AGB and key outcomes such as women’s empowerment, subordination, emotional health, and work productivity.

The following hypotheses, derived from the theoretical framework proposed, guided the empirical analysis:

**H1.** 
*Gender backlash is not exclusively violent; AGB constitutes a distinct subtype characterized by covert, non-violent, yet intentional and harmful behaviors aimed at resisting female empowerment.*


**H2.** 
*AGB is empirically and conceptually distinct from psychological violence, primarily in its motive to undermine empowerment rather than to inflict emotional harm per se.*


**H3.** 
*AGB functions as a precursor and risk factor for subsequent manifestations of IPVAW.*


**H4.** 
*AGB demonstrates higher prevalence rates than IPVAW, owing to its social normalization and lower threshold for visibility and sanction.*


**H5.** 
*AGB behaviors exist along a continuous spectrum of behavioral intensity, ranging from passive to active forms of non-violent aggression.*


**H6.** 
*AGB is a second-order latent construct composed of multiple interrelated first-order dimensions reflecting emotional, cognitive, and behavioral subcomponents.*


**H7.** 
*The underlying structure of AGB follows a semi-circumplex configuration, organizing its dimensions along axes of intensity and functional domain.*


**H8.** 
*AGB is functionally equivalent to IPVAW in its capacity to subordinate women and undermine empowerment, and comparably harmful due to its chronic, repetitive, and socially tolerated nature.*


## 2. Materials and Methods

### 2.1. Design

This study tests the conceptual model of aggressive gender reaction (AGB) using data from women members of village banks owning microenterprises in Peru. First, this research design examines the internal structure of the AGB construct. This analysis evaluates the items’ discrimination and difficulty, the construct’s second-order factor structure, and its semi-circumplex dimensionality. Second, its discriminant validity is examined, proving that the AGB is an independent construct of intimate partner violence against women (IPVAW). Finally, the predictive validity of the construct is assessed by examining the theoretically expected relationships between AGB, IPVAW, and specific dimensions of gender empowerment, including autonomy, emotional health, and work productivity.

### 2.2. Participants

The study focuses on female microentrepreneurs participating in communal banking programs, a population that is particularly relevant for examining aggressive gender backlash (AGB). These women are often beneficiaries of empowerment interventions that promote economic autonomy, leadership, and decision-making capabilities. However, such empowerment processes may also trigger resistance and backlash from intimate partners, especially in contexts where traditional gender norms remain deeply rooted. Communal banking participants represent a critical case for observing how subtle and aggressive forms of backlash can emerge in response to shifts in gender power dynamics. Moreover, this population has been widely documented in Latin American development programs, allowing for greater comparability and practical implications for violence prevention and gender policy design.

The sampling frame included all active female clients of a nationally recognized communal banking program in Peru (N = 18,106). A stratified random sampling strategy was employed to ensure geographic diversity across six regions. A total of 580 women were invited to participate, of whom 513 completed the structured interview with valid responses, resulting in an 88.4% response rate. This sample size allows for the extrapolation of results with a 95% confidence level and a 4.2% margin of error.

The surveys were conducted in 6 regions, 27 provinces, 63 districts, and all urban slums. The female respondents ranged in age from 18 to 79 years old (M = 44.8, SD = 14.5). A total of 90.4% of the respondents have between 1 and 12 children (M = 2.8, SD = 1.5). Most live with their children (79.1%), and others live with their partner (58.0%), their parents (21.9%), their siblings or nieces/nephews (14.8%), aunts/uncles or cousins (2.4%), in-laws (2.8%), or with others (18.5%). In terms of educational level, the majority have completed secondary school (47.8%), and others have incomplete secondary school (18.5%), completed university (12.1%), incomplete university (9.2%), completed primary school (7.8%), unfinished primary school (2.7%), or no education (1.8%). Regarding marital status, the majority are married (30%), and the others are cohabiting (29.4%), separated (12.9%), divorced (1.8%), widowed (6%), single without a current partner (13.1%), single with a partner (3.7%), or single without ever having had a partner (3.1%). The length of the relationship ranged from 1 to 60 years (M = 18.7, SD = 12.8).

The partners are between 18 and 80 years of age (M = 48.4, SD = 14.6). Their educational level is mainly completed secondary school (51.7%), followed by completed university (18.6%), incomplete secondary school (8.5%), completed elementary school (9%), unfinished elementary school (2.4%), and incomplete university (3.4%), or do not know (6.4%). The partners are self-employed (38.5%), dependent workers (30.2%), laborers/day laborers (8%), unemployed (7%), farmers (4%), employers or business owners (2.4%), grocers or pantry workers (2.4%), work for family (1.1%), students (1.1%), or street vendors (0.3%), among others (4.8%). Notably, 88.2% of respondents indicated that their partners received some form of payment or income for their work.

Most of them work informally. Just 21.4% of the population is registered as taxpayers. Most micro businesses are engaged in commerce (52.4%), services (43.5%), and manufacturing (9.2%), with other activities accounting for 4.1%. The main business premises are the women’s homes (73.4%), while other women are itinerant (13.2%), 8.9% have borrowed or rented premises, and only 2.8% have their own premises. More than three-quarters (77.3%) are sole proprietors of their business. A total of 22.7% have partners, mainly a partner (21.6%), or family (12.1%) or friends (12.1%). In the case of partnerships, business management is mainly shared (59.5%). Forty-five percent of women dedicate eight hours or more daily to their business. Roughly 43.2% work seven days a week in their business.

Regarding the destination of business income, nearly half (49.6%, SD = 14) is for reinvestment, while 25.7% (SD = 10.3) goes to payments, and 26.4% (SD = 13.1) is spent on non-business expenses, usually personal/family expenses. Regarding the age of the businesses, the oldest dates to 1973, while 9.4% of companies were formed in 2022. A majority (61.1%) of businesses are seven years old or younger.

In terms of loans, almost all respondents (99.2%) have requested a loan from the village bank in the last 12 months, while 13.6% have requested loans from banks or savings banks, and 0.8% have requested loans from relatives or acquaintances. Women have been village banking clients since 1992, although 14.2% obtained their first loan in 2022. A total of 67.3% have been clients for the last seven years. As for the previous loan requested, it ranges between 100 and 10,000 soles. Most respondents (95.8%) have used it for working capital, or to pay business debts (0.4%), while 3.8% used it for personal or family expenses. Only 18.2% reported having been late in paying some installment of their current credit, mainly because they became sick or had a personal emergency (30.8%), they forgot (19.8%), sales dropped (15.4%), they had personal needs (8.8%), or they were robbed (1.1%), among other reasons (27.5%). A small proportion (1.6%) reported that their partner had forced them to take out loans or had taken over the amount or installments.

### 2.3. Measures

A structured questionnaire was developed through a multi-phase and collaborative process to measure aggressive gender backlash (AGB) and its related constructs. First, a preliminary conceptual framework was elaborated based on a critical review of the literature on gender backlash, psychological aggression, and coercive control. This framework was refined through early consultations with senior researchers affiliated with the Centre for Research & Education on Violence Against Women & Children (CREVAWC) at the University of Western Ontario, whose insights helped define the construct’s theoretical boundaries.

Subsequently, an initial pool of 30 items was generated and reviewed by a technical committee composed of Peruvian experts on gender-based violence from the Movimiento Manuela Ramos, who assessed the items for content validity and cultural relevance. Cognitive interviews were then conducted with 12 female microentrepreneurs to evaluate item clarity, linguistic appropriateness, and emotional sensitivity. Finally, a pilot test was conducted with 30 participants to examine response variability and assess feasibility. As a result of this iterative process, the instrument was refined and finalized as a 21-item scale.

Data collection was carried out through face-to-face interviews administered by trained female interviewers, following standardized protocols to ensure consistency and minimize bias. Interviews were conducted in private and safe conditions, either in the participants’ homes or business premises, and informed consent was obtained from all respondents. The study protocol received ethical approval from the Ethics Committee of the Universidad de San Martín de Porres (IRB-00003251; FWA-00015320).

In addition to the AGB scale, the questionnaire included validated measures of intimate partner violence against women (IPVAW), subjective empowerment, autonomy in decision-making, emotional health, and work productivity to examine the predictive validity of the proposed construct.

Aggressive gender backlash (AGB). A 21-item reflective scale that records the partner’s aggressive reaction to women’s empowerment. The scale records hostile and instrumental aggressive behaviors. The dimensions and items have been derived from the proposed theoretical model. Women indicate the frequency of incidents on a 15-point ordinal scale: never, happened before but not now, once, or up to or more than 20 times during the last 12 months. These values are grouped as a scale, considering the sum of all items. The scale differentiates between AGB having ever occurred in the relationship and AGB during the last 12 months. The scale also distinguishes between dimensions by content and intensity levels (see Table 3).

Subjective empowerment. This 9-item scale inquires about changes in the level of empowerment during the last five years. The changes encompass education, the ability to generate and manage their finances, decision-making and problem-solving skills, their social and family influence, and their capacity to enjoy themselves. Response options range from “has worsened,” “has not improved,” remains the same, “has improved somewhat,” and “has improved significantly.” The scale’s internal consistency is good (Cronbach’s alpha = 0.805).

Autonomy. This 12-item scale measures women’s autonomy in making decisions about various personal, economic, work, and social aspects with their partner. Activities include working, studying, spending money, lending money, visiting family or friends, going to the doctor, using contraceptives, attending parties without a partner, having fun with friends, traveling, and attending meetings at the community bank. The response options are graded according to the intensity of subordination, from “does it freely”, “only informs the partner, but does not ask permission”, “informs the partner, but is not bothered”, “needs to ask permission”, or “is forbidden or does it on the sly”. The scale’s internal consistency is high (Cronbach’s alpha = 0.907).

Intimate partner violence against women. This 18-item scale examines psychological, economic, physical, and sexual violence perpetrated toward women by intimate or former partners. The items were adapted from the Conflict Tactics Scale, CTS-2 ([43]), and the survey used in the Center for Disease Control and Prevention’s National Violence Against Women Survey (NVAWS). Participants indicate the frequency of violence experienced through a 15-point ordinal scale, ranging from “never” to “more than 20 times.” The scale distinguishes between prevalence throughout the couple’s relationship and prevalence in the past 12 months. In the latter, the number of violent episodes is quantified. The scale demonstrates high internal consistency (Cronbach’s alpha = 0.905).

Emotional morbidity. This scale consists of 4 items that measure the frequency of indicators related to anxiety, depression, and suicidal thoughts during the last 12 months. Each item provides response options ranging from “never” to “more than 20 times”. The scale shows acceptable internal consistency (Cronbach’s alpha = 0.729).

Decreased labor productivity. This scale records the number of days over the past 30 days when women have experienced decreased performance due to distractions, exhaustion, mistakes in their business, or difficulties with clientele resulting from poor service. Each item provides response options ranging from “never” to “more than ten days.” The scale’s internal consistency is acceptable (Cronbach’s alpha = 0.763).

### 2.4. Analysis

The analytical strategy was structured to align with the objectives of each hypothesis. Hypotheses H1 to H4—focused on the conceptual distinction, prevalence, and relational dynamics between AGB and IPVAW—were examined using descriptive statistics, correlation analyses, and model comparisons based on exploratory and confirmatory factor analyses. Hypotheses H5 to H7, which addressed the internal structure, intensity spectrum, and latent configuration of AGB, were tested using Item Response Theory (IRT), partial least squares structural equation modeling (PLS-SEM), and polar coordinate mapping. Finally, Hypothesis H8—regarding the predictive relationship between AGB and key outcomes such as empowerment, emotional distress, and work productivity—was evaluated using correlational models. Each method was selected to rigorously test the theoretical framework and assess the psychometric robustness of the AGB scale.

The collected information was organized and tabulated in a calculation matrix, then cleaned and debugged. Data were processed using Smart-PLS 4.0, Mplus 8.0, and Python, 3.12.6 employing multiple analytical techniques—including PLS-SEM, EFA, IRT, and CFA—to validate and confirm the reliability and structure of the aggressive gender backlash construct.

PLS-SEM was used to analyze AGB’s second-order measurement model, particularly suitable given its robustness to non-normal data and its ability to avoid convergence issues more effectively than CB-SEM. First-order measurement models were examined, considering item contribution (factor loadings > 0.70), dimension reliability (composite reliability and Alpha > 0.70), convergent validity (average extracted variance, AVE > 0.50), and discriminant validity (the square root of AVE being greater than the inter-dimension correlations). Meeting these criteria confirmed that the first-order dimensions adequately represent the overall construct. Subsequently, the second-order model, representing the hierarchical relationship between dimensions and the general AGB construct, was evaluated under the same criteria.

To analyze the semi-circumplex nature of AGB, an exploratory factor analysis (EFA) with Geomin rotation was conducted, allowing for factor intercorrelations and more interpretable dimensions. Factor loadings were then visualized using polar coordinate plots in Python’s Matplotlib 3.10 library, and angular distances were calculated via the arctangent function (atan2) to test the circumplex distribution.

Item Response Theory (IRT) was applied to examine item performance, particularly relevant for constructs measuring behaviors of varying intensity. A graded response model (2PLM) was used to estimate two item parameters: discrimination and difficulty.

To test discriminant validity between AGB and external constructs, confirmatory factor analysis (CFA) was employed. Model fit was evaluated using several indices: chi-square (χ^2^), the comparative fit index (CFI), and the root mean square error of approximation (RMSEA).

Finally, to assess predictive validity, AGB dimensions were correlated with theoretically linked constructs, such as gender empowerment, subordination to male partners, emotional morbidity, and lost productive days. Consistent and significant correlations in the expected direction supported the predictive validity of the scale.

## 3. Results

### 3.1. Prevalence, Discrimination, and Item Difficulty

Table 4 shows the prevalence of indicators of AGB on five dimensions: hostility, the withdrawal of support, claims damage to masculinity, claims the loss of a caring role, and sabotage/coercion, for two time periods: during the entire relationship and in the last 12 months. The most prevalent items tend to be those theoretically classified as less intense, such as passive hostility and the withdrawal of support.

As also presented in Table 4, all item discrimination parameters exceed the value of one and are statistically significant, indicating excellent discrimination in general ([21]). Regarding the difficulty parameters, it is observed that all threshold estimators increase monotonically, as expected. Additionally, the variation in difficulty level between items is consistent with the proposed theoretical model.

### 3.2. Reliability and Construct Validity

Table 5 presents the reliability and validity analysis of the second-order construct aggressive gender backlash (AGB) and its five subconstructs. The results indicate good reliability for all constructs, with Cronbach’s alphas and composite reliabilities (Rho C) greater than 0.7. Convergent validity is evidenced by factor loadings greater than 0.5. Construct validity is generally satisfactory, with AVEs above 0.5 for AGB and most of the subconstructs, although the constructs “Sabotage/Coercion” and “Hostility” show AVEs close to the threshold.

Regarding the analysis of the semi-circumplex model, Figure 1 displays the polar coordinates generated from the rotated Geomin factor loadings, indicating a systematic distribution of the variables in a two-dimensional space, consistent with the theoretical dimensions of the AGB. The variables withdraw social support (QS), claims a loss of the caring role (EG), and claims damages to masculinity (DM) are clustered in the upper left quadrant, reflecting a greater representation of the cognitive dimension of the AGB concerning ideas about gender roles, women’s role non-compliance, and harm to male power. The variable passive hostility (HP) is located on the horizontal axis on the left side, suggesting a greater representation of passive backlash in terms of passive hostility. On the other hand, active hostility (HA) is in the upper right quadrant, indicating a strong association with active backlash. The variables sabotage (S) and coercion (C) in the upper right quadrant are more related to the active backlash, characterized by coercion and the sabotage of women’s activities. These findings support the proposed theoretical structure and provide a solid basis for interpreting the underlying dimensions of AGB in the context of the semi-circumplex model.

### 3.3. Discriminant Validity with IPVAW and Its Dimensions

To assess whether aggressive gender backlash (AGB) is a distinct construct from intimate partner violence against women (IPVAW), a series of complementary analyses were conducted.

First, prevalence comparisons (Table 6) show that AGB behaviors are more frequent than IPVAW behaviors in the sample, particularly in the last 12 months. AGB is 172.41% more prevalent than IPVAW in this period, with an average of 52.3% more incidents annually. This suggests that AGB may represent a more subtle and earlier form of resistance that can precede or coexist with overt violence.

Next, the overlap between AGB and IPVAW was examined (Table 7). While 64.5% of women who experienced AGB also reported IPVAW at some point, only 7.1% reported IPVAW without experiencing AGB. This indicates that AGB can be present independently and is not merely a byproduct of violence.

Pearson correlations (Table 8) confirm that AGB dimensions are significantly associated with psychological violence, and to a lesser extent with economic and physical violence. The magnitude of these correlations is moderate, supporting the notion that these are related but distinct constructs.

To provide a more rigorous test of discriminant validity, confirmatory factor analyses (CFA) were conducted. Table 9 presents the fit indices of various CFA models to assess the discriminant validity between the AGB and the IPVAW. The proposed models include a single-factor (1-F), a two-factor (2-F), and a two-factor model with specified correlations between dimensions within the same construct (2-F*). These fit indices are presented for annual prevalence and prevalence along the relationship. The two-factor models, 2-F and 2-F*, demonstrate substantially better fit compared to the single-factor model (1-F) for both measures, providing evidence of discriminant validity.

Finally, a parallel factor analysis (Table 10) confirms the factorial distinction between AGB and IPVAW. AGB dimensions load clearly on one factor and IPVAW on another, with minimal cross-loadings, further reinforcing the argument for discriminant validity. For both variables, the prevalence along the relationship and the annual prevalence, the dimensions of the AGB have high factor loadings on the first factor (Factor 1). On the other hand, the IPVAW dimensions have intense saturations on the second factor (Factor 2), which also supports discriminant validity. The results are consistent in the annual prevalence measurements.

Together, these findings provide robust empirical evidence of discriminant validity between AGB and IPVAW. The two constructs are conceptually related but psychometrically distinct, reinforcing the contribution of AGB as a unique and measurable form of gender-based resistance that precedes and differs from overt partner violence.

### 3.4. Predictive Validity

Table 11 presents the correlations between the dimensions of the AGB and some relevant theoretical variables. All dimensions of the AGB exhibit positive and statistically significant correlations with IPVAW, which supports their predictive validity. In addition, all dimensions of the AGB are inversely correlated with gender empowerment in the last five years, and directly associated with gender subordination towards the partner. Concerning emotional morbidity, there is a significant correlation with the dimensions of the AGB called hostility, the withdrawal of social support, and claims damages to masculinity (playing the victim). Regarding lost work productivity, positive and significant correlations are observed with all dimensions of the AGB, except for claims damages to masculinity (playing the victim).

These findings suggest a strong relationship between AGB and its dimensions, as well as with relevant theoretical variables. Specifically, women experiencing AGB tend to report higher levels of intimate partner violence, emotional morbidity, and gender subordination. Additionally, they exhibit lower work productivity and subjective feelings of empowerment. These results reinforce the predictive validity of the AGB construct.

## 4. Discussion

This study makes a significant contribution by introducing, for the first time, a comprehensive framework and a validated instrument to measure aggressive gender backlash (AGB) as a distinct construct. Previous research often relied on indirect proxies ([4]), which failed to capture the multidimensional and covert nature of AGB. By reconceptualizing AGB and validating its measurement, this study offers a robust tool for analyzing resistance to female empowerment in intimate relationships.

To synthesize the theoretical contributions of this study, Table 12 summarizes the central hypotheses of the AGB model, each addressing a distinct aspect of its nature, dynamics, and consequences.

The analysis of the data obtained provides strong support for the study’s central theoretical hypotheses, beginning with the nature and prevalence of AGB as a distinct phenomenon. The empirical results strongly support the idea that AGB is both conceptually and empirically distinct from traditional intimate partner violence against women (IPVAW). Descriptive analyses revealed that 52.1% of women reported experiencing AGB at some point in their relationship, compared to 46% who experienced IPVAW. In the past 12 months, AGB was reported at a rate 172% higher than IPVAW. These findings substantiate the hypothesis that not all forms of gender backlash are overtly violent (H1) and confirm AGB’s high prevalence and normalization (H4). This aligns with [19] ([19]) and [45] ([45]), who emphasize that resistance to gender equality often manifests in gradual and non-violent forms.

Beyond confirming AGB’s independence and predictive potential, the findings also reinforce several of the study’s core theoretical premises. Regarding the nature and prevalence of AGB (H1 and H4), the data suggest that AGB is not only more frequent than IPVAW but also potentially more insidious due to its social normalization and lack of legal sanction. That 52.1% of participants reported experiencing AGB, with an even greater differential in the past 12 months (172% more than IPVAW), illustrates its wide reach and relative invisibility within mainstream classifications of partner violence. These findings provide contemporary empirical support for classic feminist analyses of power relations ([11]; [31]), demonstrating how dominance is reinforced through both overt and covert means and operationalizing at the couple level the adverse reactions that theorists such as [17] ([17]) analyzed at the sociocultural level.

In terms of construct distinctiveness and escalation potential (H2 and H3), the study provides robust evidence differentiating AGB from psychological violence. Confirmatory and exploratory factor analyses consistently revealed that AGB and IPVAW load onto separate dimensions, affirming their psychometric independence. This distinction is based on their different motivations: AGB arises as a resistance to female empowerment and the perception of a loss of male status (a phenomenon analyzed in studies on masculinities, such as in [25]), while psychological violence may have more general motivations.

Moreover, the strong co-occurrence of AGB and IPVAW—64.5% of women who experienced IPVAW also reported AGB, compared to only 7.1% who experienced IPVAW alone—suggests a temporal or causal progression that positions the AGB as a crucial predictor and early warning sign of future violence. This escalation trajectory resonates with [41] ([41]) coercive control theory and is consistent with models of intimate partner violence that identify control as a central element preceding physical aggression (e.g., [24]).

The structural underpinnings of AGB (H5, H6, and H7) were also validated. In alignment with the General Aggression Model ([3]), the five dimensions identified—hostility, sabotage/coercion, gender stereotyping, the withdrawal of support, and masculine victimization—correspond to affective, behavioral, and cognitive mechanisms of aggression. Item Response Theory (IRT) analysis confirmed a spectrum of behavioral intensity, with lower-intensity actions such as passive hostility being more common than overtly coercive behaviors, reinforcing the notion of a graduated model (H5). The hypothesis that AGB is a second-order latent variable (H6) was supported through PLS-SEM analysis, which demonstrated a good model fit and solid convergent validity across the dimensions. Furthermore, polar coordinate analysis confirmed a semi-circumplex structure (H7), with AGB behaviors distributed along intersecting axes of intensity and internal state (emotional, cognitive, and behavioral), providing a valuable framework for visualization, training, and intervention.

Finally, the study provides compelling correlational evidence that AGB is not only pervasive but comparably harmful to IPVAW in its long-term impact on women’s lives (H8). AGB was significantly associated with diminished empowerment, elevated emotional distress, greater perceived subordination, and reduced work productivity. These findings challenge the assumption that only physical violence produces serious consequences, a finding consistent with growing evidence on the severe and lasting impact of psychological abuse on mental health, even in the absence of physical aggression ([30]; [33]).

Taken together, these findings affirm the relevance and utility of the AGB construct for advancing gender and violence research. They also offer a foundation for future empirical work and the design of interventions targeting not only overt violence but also the subtler mechanisms of gender resistance that compromise women’s autonomy and empowerment.

### 4.1. Limitations

It is important to acknowledge the limitations of this study. First, although the sample was drawn using a stratified random design, it is not nationally representative. The study focused exclusively on female microentrepreneurs participating in a specific communal banking program in Peru. While this population is theoretically relevant for exploring aggressive gender backlash (AGB), the findings may not generalize to all women, to men, or to different socioeconomic or cultural groups. Additionally, because the data were collected at a single point in time, causal inferences cannot be established. Future research using nationally representative samples and longitudinal designs is recommended to enhance generalizability and causal understanding.

Second, some dimensions of AGB yielded validity indices close to the accepted threshold. To strengthen these dimensions, the inclusion of additional items is recommended. For the active hostility dimension, the following items are proposed: “He talks to me as if he hates me,” “He does not value my economic contribution to the household,” and “He makes fun of what I earn.” For the sabotage dimension, the following items are proposed: “He does not keep his word. He tells me that he will support me, but then he goes back on his word for no reason.” For the coercion dimension, the following items are proposed: “He insists that things be as they were before, as they were when he was in charge.” The addition of these items will enable a greater depth in measuring the various facets of AGB, allowing for a more comprehensive and precise analysis of its presence and impact.

Third, while the study provides a robust conceptual and empirical framework for understanding aggressive gender backlash (AGB), it does so within a specific cultural and socioeconomic context: Peruvian female microentrepreneurs engaged in communal banking. Entrenched traditional gender norms, high levels of informality, and intense social cohesion characterize this context. These features may influence how AGB manifests and is interpreted, potentially limiting the applicability of the findings to other settings. Cultural constructs of masculinity, femininity, authority, and empowerment vary widely across societies and can shape not only behaviors but also the thresholds for what is perceived as resistance or aggression. Therefore, caution is warranted when extrapolating the findings to other populations. Future research should include systematic cross-cultural validation of the AGB scale through comparative studies in diverse sociocultural settings. This would involve testing measurement invariance across countries, adapting the scale linguistically and culturally, and examining whether the underlying dimensions of AGB retain their structure and predictive validity in different gender regimes. Such efforts would strengthen the theoretical robustness and global relevance of the construct.

### 4.2. Implications

The implications of our study are considerable and highlight the importance of measuring and understanding aggressive gender backlash (AGB) in various contexts.

(1) The prevention of IPVAW: Our findings suggest that AGB may be a precursor variable for violence. Understanding AGB and how it manifests in intimate partner relationships could help design more effective prevention interventions, allowing intervention before violence occurs. For example, community education programs could incorporate modules on the early identification of AGB, educating both men and women about subtle forms of resistance to empowerment. Furthermore, public policies need to explicitly recognize AGB as a prevalent form of aggression that precedes more serious forms of violence, creating specific strategies to identify and address this type of resistance at early stages.

On the other hand, the recognition of AGB as a mechanism that is both equally effective and as harmful as IPVAW has significant implications for both intervention and prevention strategies. First, prevention programs must expand their scope beyond physical and psychological violence to systematically include chronic patterns of gendered aggression. Second, public policy should formally recognize AGB as a priority concern, given its high prevalence, social normalization, and negative impacts on women’s decision-making, mental health, and labor outcomes. Such recognition is vital for developing integrated, gender-sensitive responses that address the full spectrum of aggression in intimate relationships.

(2) Application in refugee and migrant contexts: Although not explored in this study, future research should examine whether AGB is particularly prevalent in refugee and migrant contexts, as these families often come from countries with strongly entrenched traditional gender norms. In the host country, women may have access to new opportunities for empowerment, which may provoke gender resistance from their male partners and trigger AGB. In a society that condemns violence against women, this AGB may become the primary means for men to regain their lost sense of power. Given that these women are already facing additional challenges related to displacement and trauma, it is crucial to identify and address AGB in these contexts. This strategy can be a key component to support and reintegration efforts, helping mitigate AGB’s negative impact on migrant and refugee women and fostering their empowerment safely.

(3) Public policy: 1. The recognition of AGB’s prevalence and destructive impact provides a basis for advocating for more resources and support for victims of gender-based violence. This may include policies that explicitly address AGB as a specific form of assault far more prevalent than IPVAW and develop tailored intervention strategies. 2. The concept of AGB is crucial to deepen our understanding of the intersections between gender-based violence and women’s empowerment. Our findings suggest that AGB may act as a fundamental mechanism that impedes women’s empowerment, highlighting that public policy and intervention initiatives must address this factor to disrupt structures of gender-based violence effectively. 3. Our findings may also be relevant to understanding the disparate outcomes in economic empowerment policies, such as microcredit programs. Recognizing that AGB may be a reaction to women’s attempts at economic empowerment, policies and programs may need to consider mitigating AGB while promoting women’s economic empowerment. This critical consideration can help prevent violence and sabotage that may result from these efforts, ultimately making policies more effective and sustainable.

### 4.3. Future Research

This study opens a promising avenue for further research on AGB. Adapting and applying our measurement instrument in different contexts and populations may provide valuable insights that will enrich our understanding of AGB and its impact on society. The external validity of our findings should be considered within the specific context of the study, which focuses on women microentrepreneurs participating in community banks in Peru. While statistical representativeness is adequate for this specific population group, extrapolating these results to other groups, such as middle-class urban women or women in rural contexts without access to microcredit, should be done with caution. Future studies should assess whether the dimensions identified for AGB have a similar structure in different socioeconomic and cultural contexts.

Future research must consider how factors such as race, class, and sexuality might influence the experiences of AGB and IPVAW. Studying these intersections of identity can provide valuable insights into the diverse ways AGB is manifested and experienced. For example, indigenous women, low-income women, or LGBTIQ+ women may face specific forms of AGB that reflect the multiple forms of discrimination these populations often experience. Understanding how these intersections of identity may impact experiences of gender-based violence could provide a complete picture of the dynamics at play. It could also help develop more effective interventions and public policies tailored to the needs of different populations affected by AGB and IPVAW.

Longitudinal research is also recommended to provide a more detailed understanding of how AGB may evolve and change over time in various contexts and circumstances. A longitudinal design would allow us to empirically evaluate how behaviors associated with AGB evolve over time in response to changes in female empowerment. Specific hypotheses, such as the intensification of sabotage or the transition from passive hostility to explicit aggression following specific events of female empowerment (e.g., significant increases in income or economic independence), could be rigorously investigated using this design.

Future research could explore how men experience or rationalize AGB behaviors, particularly concerning evolving gender norms. While not examined in this study, such exploration may enrich the understanding of resistance mechanisms. Qualitative studies employing in-depth interviews or focus groups could be useful in capturing detailed and contextualized accounts of AGB from the male perspective. These studies may offer insights into the motivations, justifications, and cognitive–emotional processes associated with AGB. To structure such inquiry, the General Aggression Model (GAM) could serve as a useful theoretical lens. GAM conceptualizes aggression as the outcome of interactions between personal and situational factors, including the individual’s cognitive interpretation of the context, emotional state, and physiological arousal. Applying this model may help elucidate how these elements jointly shape aggressive backlash behaviors in intimate contexts. Research of this kind could contribute to a broader and more nuanced understanding of AGB, potentially informing the development of more effective prevention strategies that address its underlying drivers and help mitigate its escalation toward IPVAW.

## 5. Conclusions

This study offers the first comprehensive theoretical and empirical model of aggressive gender backlash (AGB) in intimate partner relationships, establishing it as a distinct, measurable, and impactful phenomenon. By developing and validating a multidimensional scale grounded in the General Aggression Model, this study provides a reliable instrument for identifying subtle but systematic forms of resistance to female empowerment. The findings reveal that AGB is not only more prevalent than intimate partner violence but also often precedes it, and may function as a chronic mechanism of subordination, as suggested by its strong associations with emotional and economic consequences. Crucially, AGB operates in ways that are socially tolerated and legally unrecognized, which increases its invisibility and persistence. Recognizing AGB as a unique and harmful expression of gendered aggression has important implications for public policy, prevention programs, and gender equality interventions. Future research should extend this framework to diverse populations and sociocultural contexts, explore its long-term effects, and incorporate male perspectives further to elucidate the cognitive and emotional mechanisms underlying resistance. Addressing AGB is not only key to preventing violence—it is essential to dismantling the structural barriers that continue to undermine women’s autonomy and well-being.

## Figures and Tables

**Figure 1 behavsci-15-00941-f001:**
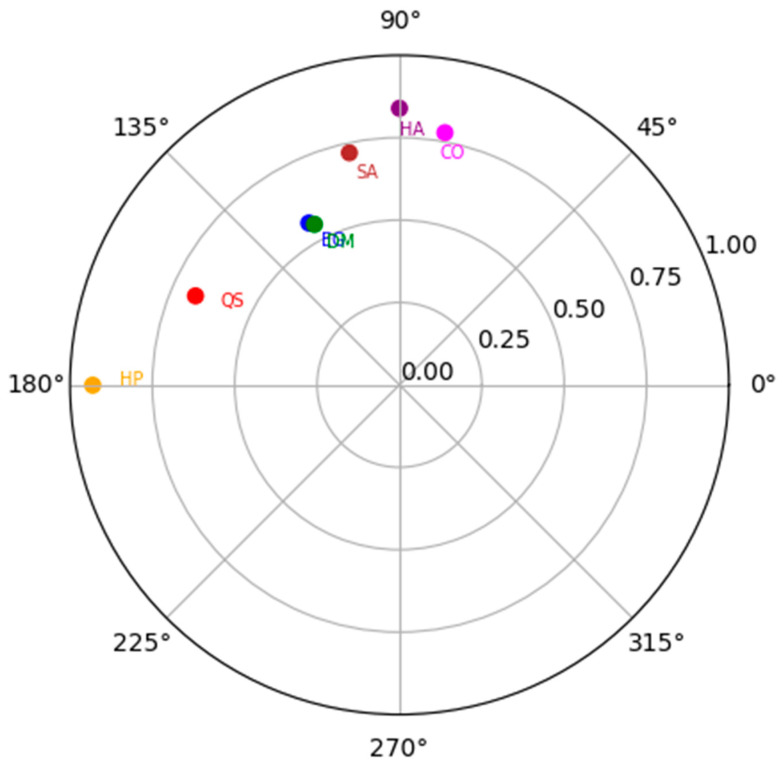
Semi-circumplex model of aggressive gender backlash. Note. The exact coordinates of each variable were retained to preserve the model’s analytical structure. QS = Withdraw Social Support; EG = Claims Loss of the Caring Role; DM = Claims Damages to Masculinity; HP = Passive Hostility; HA = Active Hostility; S = Sabotage; C = Coercion. Radial coordinates constructed with the standardized factor loadings from factor analysis with Geomin rotation. RMSEA = 0.067, *p* = 0.203; SRMR = 0.017; CFI = 0.991; TLI = 0.975; and χ^2^ (df) = 19.4 (8), *p* = 0.012.

**Table 1 behavsci-15-00941-t001:** Key differences between aggressive gender backlash (AGB) and traditional psychological violence.

Aspect	Traditional Psychological Violence	Aggressive Gender Backlash (AGB)
Primary Objective	To cause direct emotional harm, erode self-esteem, and control through fear or explicit humiliation.	Specific resistance to women’s empowerment through indirect actions aimed at reversing or hindering changes in gender power dynamics.
Common Manifestations	Insults, explicit threats, humiliation, constant monitoring, direct emotional blackmail, and verbal aggression.	Subtle withdrawal of emotional support, indirect sabotage of personal or work goals, indirect recriminations, and blaming for neglecting traditional gender roles.
Explicitness Level	Explicit, easily identifiable as aggression or violence.	More subtle, often mistaken for everyday conflicts or disagreements; less socially recognizable as aggression.
Social Consequences	Lower social tolerance, higher likelihood of being reported, and easily recognized by third parties as unacceptable behavior.	Higher social tolerance, perceived as less severe, and rarely reported or recognized as violent behavior.
Legal Risk	Higher risk of legal sanctions or judicial interventions due to formal complaints.	Low risk of legal consequences due to difficulty proving or legally recognizing it as aggression.
Effects on the Victim	Deep emotional harm, and severe psychological disorders such as anxiety, chronic depression, and evident psychological trauma.	Significant negative emotional impact, progressive deterioration of autonomy and self-esteem specifically related to empowerment, and frustration of personal and professional goals.
Role in Relationship	Clear domination, direct emotional control over the partner, and explicit maintenance of power through intimidation.	Indirect maintenance of power through subtle strategies; specific resistance to changes in traditional gender status without necessarily seeking explicit domination through direct fear or intimidation.

**Table 2 behavsci-15-00941-t002:** Application of the General Model of Aggression (proximal process) to AGB.

Processes	Dimensions	Features	How Would Apply to AGB?
Personal and situational input	Aspects of the situation	Frustration Challenging provocation Social stress Macho socialization Aggressive social models Gender stereotypes Ideal “family” models	Empowered women stop assuming traditional gender roles, increasing men’s frustration. Men socialized with traditional gender roles, with hegemonic masculinity, are more likely to feel frustrated or challenged by their partners’ new roles.
	Aspects of the individual	Impulsivity Narcissism/Neuroticism Unstable self-esteem Aggressive self-image Male chauvinist attitudes Shifting of responsibility	Men with unstable self-esteem and tolerant attitudes and beliefs towards IPVAW are more prone to AGB.
Internal routes	Physiological activation	Arousal (excitement)	Frustration is felt as intense displeasure, which can increase when hostile feelings or irrational beliefs feed it.
	Affective states	Hostile feelings Ira Indignation	They are automatic, mainly when men feel socially or personally devalued. Men feel anger at “women’s lack of respect”.
	Cognitive states	Irrational beliefs Hostile thoughts Cognitive distortions (attribution biases)	Believing that gender subordination is normal. Blaming the partner for the “insubordination”. Blaming the partner for frustration (false attribution). Believing that the partner is being unfair to him (false justice).
Results	Evaluation and decision process	The cost–benefit evaluation	Cost of the social/legal sanction Benefit of the objective achieved
	Impulsive action (hostility)	Chronic hostility	(Graduated by intensity.) (1) Annoyance. (2) Torturing silence. (3) Recrimination. (4) Minimization. (5) Aggressive transfer. (6) Aggressive treatment.
	Thoughtful action (instrumental)	Gender manipulation	(1) Reduction of social support. (2) Stereotyping. (3) Manipulation. (4) Blame. (5) Sabotage. (6) Repression.
Social gathering	Feedback	Social/legal sanction Subordination achieved Intensity increase	

Note. Authors’ own elaboration based on [3] ([3]). IPVAW = intimate partner violence against women. AGB = aggressive gender backlash.

**Table 3 behavsci-15-00941-t003:** Aggressive gender backlash: content, dimensions, and indicators.

Content/Dimension/Description	Indicators
Emotional	
Hostility: Refers to expressing negative emotions, such as anger, resentment, and contempt, towards the partner. It includes annoyance, scolding, glaring, or belittling the partner’s achievements. It has two levels of intensity (passive–active).	Passive: Has shown annoyance, or bothered you, without telling you the reason. Scolds you for any mistake you make. Blames you for the things you do. Looks at you angrily, as if despising or wanting to hit you. Active: Makes you feel like your goals (work/study) are worthless. Mocks your personal/professional achievements. Mistreats your belongings, as if wanting to destroy them.
Behavioral	
Withdraw support: Addresses the withdrawal of emotional and social support by the partner, manifested through evasive behaviors. It has one level of intensity.	Treats you coldly, without affection or warmth. Has stopped talking to you, does not look at you, ignores you/acts as if you do not exist for him. Has withdrawn their support as a partner. Does not support you when you ask for it. Refuses to support you with your goals.
Sabotage/Coercion: Focuses on instrumental behaviors that seek to hinder or complicate the achievement of the personal and professional goals of the partner. It has two levels of intensity (sabotage and coercion).	Sabotage: Pretends to support you but puts obstacles in your way. Deliberately makes things difficult for you. Complicates your life. Coercion: Does everything possible to make you abandon your goals. Pressures you to quit working/studying.
Cognitive	
Claims for loss of gender role (stereotyping): Refers to behaviors through which the partner blames the woman for not fulfilling traditional roles of motherhood or wifehood, reflecting internalized gender stereotypes. It has one level of intensity.	Complains that you fulfill your role as a mother or wife. Tells you that you are neglecting your roles as a mother or wife. Tells you that because of your work or studies, you are neglecting the home, the children, or him. Makes you feel bad, that you are not a good mother or wife. Blames you for the problems in the house. Blames you for his infidelities. Says you have disappointed him, that you are not the woman/wife he expected.
Claims for affecting his masculinity (playing the victim): Refers to verbal behaviors in which the partner blames the woman for undermining his authority, status, or image as a man. These expressions reflect an attempt to portray himself as a victim and restore perceived masculine dominance. It has one level of intensity.	Says you took away his authority in front of your children. Claims you no longer value him as the head of the household. Accused you of making him look bad in front of his friends/family. Claims you no longer respect him as a man. Says you make him feel less/worthless.

**Table 4 behavsci-15-00941-t004:** Prevalence of indicators of aggressive gender backlash (ordered by dimension), and level of discrimination and difficulty using Item Response Theory.

Dimensions/Items	Prevalence		Difficulty and Discrimination			
	During the Relationship	Last 12 Months	Diff (1)	Diff (2)	Disc	(Z)
Hostility	47.1	27.6				
Shows annoyance or anger without explanation.	36.5	19.2	0.505	1.152	2.161	7.73
Scolds or blames you for any mistake.	32.1	18.8	0.650	1.134	2.487	7.59
Looks at you with contempt or aggression.	18.8	6.2	1.032	1.61	3.849	5.24
Devalues your goals (e.g., work/study) or mocks your achievements.	12.0	5.6	1.269	1.591	4.720	5.21
Damages or mistreats your belongings with intent.	6.8	1.2	1.540	2.150	4.118	4.55
Withdraw social support	30.3	12.7				
Stops talking, avoids eye contact, or acts as if you do not exist.	22.8	9.0	0.928	1.554	2.676	7.02
Treats you coldly, without affection or emotional warmth.	21.6	7.4	0.976	1.622	2.937	6.82
Refuses support when requested or withdraws it deliberately.	10.5	3.4	1.340	1.786	4.255	5.12
Claims damages to masculinity (playing the victim)	21.8	9.5				
Says you undermine his authority in front of your children.	14.8	5.5	1.343	1.925	2.369	5.92
Claims you no longer value him as head of the household.	13.9	4.0	1.310	1.962	2.865	5.85
Claims you no longer respect him as a man.	10.5	2.8	1.358	1.909	3.912	5.21
Says you make him look bad in front of his friends/family.	9.0	3.1	1.450	1.906	4.079	5.03
Claims you make him feel less/inferior/disrespected.	8.6	1.8	1.406	1.952	4.908	4.78
Claims loss of caring role (stereotyping)	23.8	9.3				
Criticizes you for neglecting household, children, or him due to work/study. Makes you feel like a bad mother or wife.	15.5	6.2	1.307	1.894	2.255	6.03
Demands compliance with traditional gender roles.	14.2	5.6	1.355	1.924	2.342	5.91
Blames you for domestic problems or his own infidelities.	10.2	3.4	1.412	1.919	3.25	5.42
Expresses disappointment that you are not the woman/wife he expected.	10.5	2.8	1.344	1.869	4.051	5.25
Sabotage/Coercion	20.7	7.1				
Creates obstacles deliberately or makes your life harder.	14.5	3.4	1.224	1.914	3.326	5.92
Pressures you to quit working/studying.	9.9	3.1	1.661	2.33	2.142	5.11
Pretends to support you but puts obstacles in your way.	8.0	2.8	1.478	1.882	4.257	4.79
Does everything possible to make you abandon your goals.	7.7	1.5	1.616	2.291	2.960	4.74

Note. Prevalence refers to the percentage of women who report each AGB indicator during the relationship or within the past 12 months. Diff (1) and Diff (2) represent item difficulty for reporting the behavior once or more than once, respectively. Disc indicates item Discrimination—its ability to differentiate between levels of AGB exposure. (Z) is the significance (z-score) of the discrimination estimate.

**Table 5 behavsci-15-00941-t005:** Reliability and second-order construct validity of the AGB.

	Items	Factor Loadings (Range)	Cronbach’s Alpha	Composite Reliability (Rho C)	Average Variance Extracted (AVE)
**Second**-**order construct**					
Aggressive gender backlash	5	0.787–0.909	0.924	0.933	0.678
**First-order constructs**					
Claims loss of caring role (stereotyping)	4	0.705–0.845	0.776	0.857	0.601
Claims damages to masculinity (playing the victim)	5	0.678–0.863	0.844	0.890	0.619
Sabotage/Coercion	4	0.699–0.783	0.736	0.833	0.556
Withdraw social support	3	0.795–0.838	0.738	0.851	0.656
Hostility	5	0.644–0.811	0.781	0.852	0.536

**Table 6 behavsci-15-00941-t006:** Percentage differences between IPVAW and AGB.

	IPVAW	AGB	Difference (%)
Ever in the relationship	46.00	52.10	13.26
In the last 12 months	11.60	31.60	172.41
Average incidents (last 12 months)	27.10	41.30	52.39

**Table 7 behavsci-15-00941-t007:** Relationship between AGB and IPVAW at the prevalence level.

Intimate Partner Violence Against Women	AGB During the Relationship ^a^		AGB During the Last 12 Months ^b^	
	No	Yes	No	Yes
No	92.9%	35.5%	97.3%	70.9%
Yes	7.1%	64.5%	2.7%	29.1%

Note. ^a^ Pearson’s χ^2^ = 114.23, *p* < 0.001; ^b^ Pearson’s χ^2^ = 49.62, *p* < 0.001.

**Table 8 behavsci-15-00941-t008:** Pearson’s correlations between AGB dimensions and IPVAW over the last 12 months.

	Psychological	Economic	Physical	Sexual
Hostility	0.429 **	0.135 **	0.151 **	0.142 **
Withdraw social support	0.389 **	0.254 **	0.112 **	0.057 **
Claims loss of caring role (stereotyping)	0.347 **	0.125 **	0.100 **	0.195 **
Sabotage/Coercion	0.323 **	0.107 **	0.093 **	−0.010
Claims damages to masculinity (playing the victim)	0.339 **	0.145 **	0.167 **	0.052 **

Note. ** *p* < 0.001.

**Table 9 behavsci-15-00941-t009:** Discriminant validity between AGB and IPVAW.

Fit Indices	Prevalence During the Last 12 Months			Prevalence During the Relationship		
	1-F	2-F	2-F *	1-F	2-F	2-F *
χ^2^ (df)	254 (27)	100.63 (26)	83.38 (25)	150.48 (27)	82.0 (26)	47.51 (24)
CFI	0.823	0.942	0.955	0.941	0.973	0.989
TLI	0.764	0.920	0.935	0.921	0.963	0.983
NFI	0.807	0.924	0.937	0.929	0.961	0.978
GFI	0.673	0.944	0.954	0.689	0.894	0.890
RMSEA	0.131	0.076	0.069	0.096	0.066	0.045
SRMR	0.183	0.127	0.128	0.050	0.046	0.028

Note. Maximum likelihood estimates. * Factors with correlated errors between some dimensions of the same construct, according to theory. χ^2^ (df) = Chi-square statistic with degrees of freedom; CFI = Comparative Fit Index; TLI = Tucker–Lewis Index; NFI = Normed Fit Index; GFI = Goodness of Fit Index; RMSEA = Root Mean Square Error of Approximation; SRMR = Standardized Root Mean Square Residual.

**Table 10 behavsci-15-00941-t010:** Parallel factor analysis between the dimensions of the AGB and the IPVAW.

Dimensions	Factor 1	Factor 2	Uniqueness
Prevalence during the relationship			
Psychological violence		0.785	0.180
Economic violence		0.770	0.383
Physical violence		0.677	0.357
Sexual violence		0.594	0.712
Hostility	0.936		0.149
Withdraw social support	0.869		0.320
Claims loss of caring role (stereotyping)	0.666		0.498
Sabotage/Coercion	0.675		0.440
Claims damages to masculinity (playing the victim)	0.432		0.482
Prevalence during the last 12 months			
Psychological violence	0.207	0.840	0.108
Economic violence		0.656	0.603
Physical violence		0.655	0.575
Sexual violence		0.445	0.809
Hostility	0.860		0.268
Withdraw social support	0.754		0.432
Claims loss of caring role (stereotyping)	0.630		0.556
Sabotage/Coercion	0.662		0.596
Claims damages to masculinity (playing the victim)	0.701		0.505

Note. Extraction of factors through parallel and minimum residual analysis. Promax rotation. Factor loadings less than 0.2 were eliminated.

**Table 11 behavsci-15-00941-t011:** Correlations between AGB and its dimensions (last 12 months), with some theoretically related variables.

AGB	IPVAW	IPVAW (12 Months)	GEP	GSU	EMB	LLP
Hostility	0.520 **	0.373 **	−0.236 **	0.305 **	0.214 **	0.181 **
Withdraw social support	0.542 **	0.353 **	−0.181 **	0.230 **	0.118 *	0.122 *
Claims loss of caring role (stereotyping)	0.426 **	0.341 **	−0.001	0.219 **	0.067	0.078
Sabotage/Coercion	0.424 **	0.260 **	−0.252 **	0.258 **	0.099	0.141 *
Claims damages to masculinity (playing the victim)	0.491 **	0.337 **	−0.166 **	0.338 **	0.111 *	0.130 *
AGB (last 12 months prevalence)	0.583 **	0.418 **	−0.213 **	0.345 **	0.176 **	0.238 **
AGB (relationship prevalence)	0.690 **	0.381 **	−0.198 **	0.291 **	0.170 **	0.192 **

Note. GEP = gender empowerment in the last five years. GSU = gender subordination to partner. EMB = emotional morbidity. LLP = lost labor productivity. ** *p* < 0.001, * *p* < 0.05.

**Table 12 behavsci-15-00941-t012:** Summary of contrasted theoretical hypotheses on aggressive gender backlash (AGB).

Hypothesis/Statement	Theoretical Contribution	Implications	Empirical Conclusion/Evidence
H1. Not all gender backlash is violent.	Expands GB beyond physical/emotional violence; defines AGB as non-violent but aggressive.	Allows AGB to be measured as an independent variable.	Descriptive evidence: 52.1% of women reported AGB vs. 46% reporting IPVAW. This suggests AGB can occur in the absence of physical violence, though a formal test of difference was not conducted.
H2. AGB is different from psychological violence.	Introduces distinct motivational and functional features (resistance vs. harm).	Justify the creation of a specific scale.	Confirmed via discriminant validity: CFA and EFA indicated distinct latent structures between AGB and psychological violence.
H3. AGB is a precursor and risk factor for IPVAW.	AGB may precede or escalate into IPVAW under certain conditions.	AGB should be targeted in early prevention efforts.	Correlational evidence: 64.5% of those experiencing AGB also reported IPVAW; only 7.1% reported IPVAW without AGB. While suggestive, no longitudinal or predictive model was tested.
H4. AGB is more prevalent than IPVAW.	Highlights the normalized and underestimated presence of AGB.	AGB must be addressed in public policy.	Descriptive evidence: 172% higher prevalence of AGB than IPVAW in the last 12 months. No inferential test of proportion difference was conducted.
H5. AGB has a graduated behavioral intensity spectrum.	Frames AGB behaviors along an intensity continuum.	Facilitates the construction of graded scales and risk profiles.	Supported by IRT analysis: Items showed graded difficulty and prevalence levels; lower-intensity items were more frequent than high-intensity ones.
H6. AGB is a second-order latent variable.	Positions AGB as a multidimensional construct.	Justifies the use of SEM and scales with multiple dimensions.	Confirmed: SEM-PLS analysis demonstrated adequate model fit for five dimensions with good convergent validity.
H7. AGB has a semi-circumplex structure.	Suggests spatial and functional interrelation between AGB dimensions.	Supports visual models for training and intervention.	Supported: Polar coordinate analysis and factor rotation showed a structured two-dimensional pattern; CFI = 0.991.
H8. AGB is as effective as IPVAW in subordinating women and limiting their empowerment, and as harmful due to its chronic nature.	Proposes functional and affective equivalence between AGB and IPVAW. Challenges the assumption that only violence produces serious harm.	Interventions and public policy should treat AGB as a high-priority mechanism of subordination.	Correlational evidence: AGB is associated with lower empowerment, emotional distress, and lower productivity. Causality or equivalence not formally tested.

## Data Availability

The original data presented in the study are openly available on https://osf.io/2jex3/?view_only=644b5c273d60443eb9b4de85c40e599b (accessed on 13 May 2025).

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
