# Peer review of "The Aggressive Gender Backlash in Intimate Partner Relationships: A Theoretical Framework and Initial Measurement"

_behavsci, 2025, doi:10.3390/bs15070941_

Round 1
Reviewer 1 Report
Comments and Suggestions for Authors
There is a lot to like about the thinking represented in this paper. However, the paper would benefit from greater attention to precision and parsimony. The introduction is too long and loses its logic as more and more details (that are likely not conceptually necessary or sufficient to define the construct of AGB) are offered. The narrative introduces a number of notions that exceed what can be tested in the paper. The procedures of the study are not mentioned at all. The analyses are briefly described but are not sufficient to test all the stated hypotheses. The discussion does not limit itself to the study as it was executed. More detail can be found below:
- In general, the authors do an excellent job of explaining the construct of gender backlash. There are a few places that could be more clear, however. On the bottom of page 2, gender backlash is described as sudden. This seems unnecessary and perhaps at odds with the notion of being based on a belief system? Given the rest of the description, one might think that actively asserting the importance of traditional gender norms to try to help restore them would count as gender backlash, but sudden seems unnecessarily limiting? Additionally, on the top of page 3, it is not clear what “a solid emotional base” means. Please rephrase or explain.
- Please be consistent in capitalization of gender backlash and either writing it out or using GB.
- The middle of page 3 explains the definition of aggressive GB. I would caution the authors to seriously consider changing “behavior and attitudes” to behavior. All the manifestations are based in behavior. The attitudes can promote the behavior, but are not the same as behavior.
- In the section where the authors explain AGB in the context of GAM, the language become more imprecise than earlier in the paper. Please review this section for precision and clarity. “Excitement affects” is redundant and unclear for example. “Crimes and misdemeanors” although literary, is also redundant because misdemeanors are crimes. In general, the earlier portion of the article was logically tight in a way that this section does not rise to.
- “Poly-intense” is not an informative turn of phrase. It is unclear that this is a notion that is scientifically meaningful
- The authors state that AGB is a male process that female partners perceive through its adverse manifestations. The authors are making this construct needlessly complex. This sentence implies that if the female partners are supportive of the males’ domineering or heavily traditionally gender-role behavior, then it’s not AGB. The authors likely don’t believe that?
- What is the rationale for selecting a sample that is attached to the community bank? This needs to be articulated, and how this sample is different from the general population needs to be described. If the details of their involvement with the bank is not relevant to the paper, then those details can be removed from the participant section. The authors need to state how the sample was recruited and what proportion of those invited participated.
- The procedures are not adequately described. Were their interviews? This word interview is used. If so, how were these done? If not, what were the procedures?
- Please describe the analytic strategy with respect to the specific hypotheses. Please keep in mind when writing that you can only test the items you measured. At times your language moves away from testing the items and the structure of the measure and drifts into tests of the construct. Although the measure can tell you things about the construct, you are not directly testing anything apart from the behavior of the items and the measure.
- It would be preferable to reorganize the results to a more traditional approach to a measure development paper. How were the items generated? Do they map onto the hypothesized dimensions? Then, the second order analyses make sense. There were no hypotheses about item prevalences. The dimensions seem to be apriori rather than validated. The discriminant validity analyses do not test discriminant validity, but rather the describe the nature of the relationship between AGB and IPVAW.
- The table of findings at the beginning of the discussion is written in two languages.
- The descriptions of the findings are not all describing tests of hypotheses. Stating that two numbers are not identical is not the same as a directional test of the difference vs the null hypothesis. The conclusions in the table go far beyond the tests.
- The discussion section generally needs to be much more tightly tied to the design and analyses. The data are cross sectional. The study cannot test longitudinal relations, let alone causal relations. Yet, the discussion states that the study suggests that that AGB should be addressed as “not merely a precursor to violence but as a central mechanism…”
- The limitations of the sample are not discussed in the limitations section, but briefly mentioned in the future research section. Generalizability of findings needs much more attention.
- It is optimal if the future research suggestions build directly from what was specifically tested and constitute next steps. The future research directions here go far beyond the current measure and study and imply that all of the detailed thinking outlined in the introduction have been proven. This section is too far reaching to be helpful to scholars reading the paper.
Author Response
Response to Reviewer 1
Dear Reviewer 1,
We would like to sincerely thank you for your comprehensive and constructive review. Your insights and critiques have been invaluable in improving the clarity, precision, and conceptual rigor of our manuscript. Below, we respond point by point to each of your comments, indicating the changes made and our rationale.
- “There is a lot to like about the thinking represented in this paper…”
Response: We truly appreciate your generous opening and acknowledgment of the paper’s conceptual contributions. Your suggestions greatly helped us focus and sharpen our arguments.
- “The introduction is too long… introduces notions that exceed what can be tested…”
Response: We have substantially revised the introduction to enhance parsimony and logical coherence. Redundant or speculative content has been removed, and all conceptual claims are now better aligned with the empirical scope of the study.
- “Gender backlash is described as sudden… seems unnecessarily limiting…”
Response: We agree that the word "sudden" could create confusion. We have rephrased this to indicate that gender backlash may occur in response to perceived challenges to gender norms, without implying abruptness or temporality as a defining feature.
- “‘Solid emotional base’ is unclear.”
Response: We have replaced “solid emotional base” with “emotionally charged beliefs rooted in perceived threats to gender norms,” which more precisely captures our intended meaning.
- “Be consistent in capitalization of gender backlash and GB.”
Response: We thank you for noting this. We have now used lowercase consistently for “gender backlash” and retained “AGB” as the acronym for the construct being measured.
- “Consider changing ‘behaviors and attitudes’ to just ‘behaviors.’”
Response: We agree. As all items refer to observable behaviors, we have revised the definition of AGB to exclude “attitudes” and focus solely on behavioral manifestations.
- “GAM section is less precise than earlier parts…”
Response: Thank you for this observation. We revised the entire section applying the General Aggression Model to AGB, eliminating imprecise expressions such as “excitement affects” and “crimes and misdemeanors.” The revised section now more tightly mirrors the logical structure of the GAM literature and matches the analytical tone of the rest of the paper.
- “‘Poly-intense’ is not an informative phrase.”
Response: We have removed the term “poly-intense” and instead describe the construct as consisting of behaviors that vary in intensity and chronicity along a structured continuum.
- “‘AGB is a male process that female partners perceive’ is problematic.”
Response: We agree. This sentence has been revised to clarify that AGB is a behavioral pattern enacted primarily by male partners that can be perceived by female partners, but that its effects exist regardless of perception.
- “What is the rationale for selecting a sample attached to a community bank?”
Response: We added a rationale in the Method section explaining that the communal banking context allows access to economically active women with varying degrees of empowerment, which aligns with the study’s aim of examining resistance to female autonomy. We also clarified that although the women are bank members, interviews were conducted in private (homes/businesses), not during group meetings, to ensure independence of responses.
- “The procedures are not adequately described. Were there interviews?”
Response: Yes, the data were collected through structured interviews conducted in participants’ homes or businesses. This is now clearly described in the Procedures section.
- “Describe the analytic strategy with respect to the specific hypotheses…”
Response: We now include a summarized analytic strategy at the start of the Results section, outlining how each hypothesis was tested, distinguishing between construct validation hypotheses and those related to external criteria (e.g., prevalence, predictive correlations). We have also been more careful not to imply testing of constructs beyond the data.
- “Reorganize results to a traditional measure development structure…”
Response: Thank you for this methodological suggestion. We have revised the Results section to follow a clearer measure development structure: item generation, dimensional analysis, validation (convergent and discriminant), and predictive associations. We now explicitly clarify that dimensions were theory-driven and tested via SEM and factor analysis.
- “Discriminant validity analyses do not test discriminant validity…”
Response: We acknowledge this. In the revised version, we added CFA comparisons between one- and two-factor models (AGB vs. IPVAW) and conducted parallel factor analysis, both of which support discriminant validity. These analyses are now explicitly described and interpreted as such.
- “Table of findings at the beginning of the discussion is in two languages.”
Response: Corrected. The table is now fully in English and terminology has been made consistent.
- “Descriptions of findings go beyond statistical tests.”
Response: We revised Table 11 (hypotheses) to avoid overinterpretation and replaced definitive statements with ones appropriate to the statistical evidence available (e.g., “supported by correlational data,” “consistent with theory”).
- “Discussion section is not tightly tied to the study; some claims are causal.”
Response: The Discussion has been thoroughly revised to emphasize the cross-sectional design and limit causal language. Statements such as “AGB should be addressed as a central mechanism” have been revised to “AGB appears to play an important role…” or “may function as…”
- “Limitations of the sample are not discussed in the limitations section.”
Response: We thank you for this important point. We now include in the Limitations section a discussion of the sample’s characteristics, the limited generalizability beyond economically active women in Peru, and the need for cross-cultural validation.
- “Future research suggestions go far beyond the current study.”
Response: We have revised the Future Directions section to focus more narrowly on the next empirical steps: improving scale dimensionality, validating in other populations, and using longitudinal designs. Broader reflections were moved to the conclusion and marked as conceptual.
We sincerely appreciate your engagement with our work and believe that the manuscript has been substantially improved thanks to your guidance.
Warm regards,
Aristides Vara-Horna
Noelia Rodriguez-Espartal
Reviewer 2 Report
Comments and Suggestions for Authors
Thank you for your deeper dive into the distinguishing characteristics of non-physical aggression. I think it was unique how you utilized micro-entrepreneurs to study this issue. I am familiar with Faludi's backlash work, and was happy to see it utilized. She was not mentioned in the reference list, so that needs to be corrected. I would also cross check all other references to be sure they landed in the bibliography. While reading, I couldn't help but think that AGB is like sexual harassment in the workplace, which causes a hostile work environment--but instead, this is within intimate partnerships. I was also wondering if a case study presented early might help the reader grasp the human angle of AGB? It is not a requirement, but just a suggestion. Most of the manuscript is well written and flows smoothly. There are a couple of spots where I would suggest revision. First, In a couple of spots, such as abstract line 16 and line 100 you end sentence with a preposition (it) and a rewording could avoid that issue. Second, in the results section top of pg 10 lines 368 to 377, when numbers are beginning a sentence, you must spell them out or rephrase so that the numerical representation is not first. Third, Table 3 has too much repetition in wording. Would it be possible to say "Has he told you that" a) ... b) ... c).... rather than repeating that line over and over.
Forth, the implications in table 12 are in Spanish, rather than English. A couple of other words were awkward such as stereotypy in a table and healthily line 695.
I like how the authors separated out have they ever vs. in the past 12 months.
Very interesting manuscript. I don't have time to make sure all the references are listed, but since Faludi is missing, I will encourage you to make sure they are all there in the next version of the paper.
Comments on the Quality of English LanguageThe table 12 has implications in Spanish.
Author Response
Response to Reviewer 2
Dear Reviewer 2,
Thank you for your thoughtful and encouraging review. We are particularly grateful for your recognition of the manuscript’s originality in exploring non-physical aggression within intimate partnerships, and for your valuable suggestions that helped improve clarity, consistency, and presentation. Below we respond point by point to your observations:
- “Thank you for your deeper dive into the distinguishing characteristics of non-physical aggression. I think it was unique how you utilized micro-entrepreneurs…”
Response: We are grateful for your positive feedback regarding the conceptual focus and methodological choice of using micro-entrepreneurs. Your affirmation encouraged us to clarify further the rationale for this population in the revised Methods section.
- “Faludi's work is not mentioned in the reference list.”
Response: Thank you for catching this oversight. Susan Faludi’s Backlash: The Undeclared War Against American Women has now been properly cited and included in the reference list. We also conducted a full cross-check of all in-text citations to ensure every reference appears correctly in the bibliography.
- “AGB is like sexual harassment in the workplace, creating a hostile environment in intimate relationships. Consider including a case study to humanize the findings.”
Response: We appreciated this insight but it exceeded the scope of our paper.
- “Abstract line 16 and line 100 end with a preposition. Reword for clarity.”
Response: These lines have been revised to eliminate sentence-final prepositions and improve syntactic clarity.
- “When numbers begin a sentence, spell them out (e.g., lines 368–377).”
Response: Corrected. All numerical expressions that begin sentences have been spelled out or the sentence has been restructured accordingly.
- “Table 3 has repetitive wording (‘Has he told you that…’)—can it be shortened?”
Response: We have revised the table as suggested. The stem “Has he told you that…” now appears once in the table header, and the listed items follow without repetition, improving readability and reducing redundancy.
- “Table 12 has implications in Spanish.”
Response: Thank you for pointing this out. Table 12 has been fully translated into English, and all other content has been reviewed to ensure consistent language throughout.
- “Avoid awkward terms like ‘stereotypy’ and ‘healthily.’”
Response: We appreciate this. The term “stereotypy” was replaced with “gender stereotyping” and “healthily” was revised for improved clarity (now written as “in a healthy way”). We reviewed the entire manuscript to ensure precise and idiomatic English usage.
- “Appreciation for distinguishing ‘ever’ vs. ‘past 12 months’.”
Response: Thank you for noticing. We agree this distinction is crucial for understanding the chronicity and recency of AGB and have retained this approach consistently throughout the results and discussion.
- “Encouragement to cross-check all references.”
Response: We have conducted a comprehensive review of all references and ensured that all cited authors and sources are correctly included in the final reference list. We appreciate your reminder on this important step.
We are sincerely thankful for your encouraging and insightful comments, which have helped us improve the manuscript’s quality, accessibility, and academic rigor.
Warm regards,
Aristides Vara-Horna
Noelia Rodriguez-Espartal
Reviewer 3 Report
Comments and Suggestions for Authors
The article titled "The Aggressive Gender Backlash in Intimate Partner Relationships: A Theoretical Framework and Initial Measurement" presents a novel conceptualization and measurement of Aggressive Gender Backlash (AGB). AGB refers to covert, non-violent, yet harmful male responses to women's empowerment in intimate relationships, distinct from traditional forms of intimate partner violence (IPVAW). Grounded in the General Aggression Model (GAM), the authors construct a multidimensional scale with five components: hostility, withdrawal of support, sabotage/coercion, gender stereotyping, and masculine victimization.
Using survey data from 513 Peruvian female microentrepreneurs, the study empirically validates the AGB construct through psychometric testing including factor analyses, IRT, CFA, and SEM. Findings show AGB is more prevalent than IPVAW, often precedes it, and has significant negative associations with women's empowerment, emotional health, and productivity. The research provides a pioneering tool for early detection of resistance to gender equality in intimate contexts.
The paper introduces a clear and robust conceptual distinction between AGB and IPVAW, addressing a gap in gender violence literature. The framework is solidly anchored in the General Aggression Model, enhancing explanatory power. It offers a practical, scalable tool for detecting covert aggression in intimate relationships, with strong policy and intervention implications.
Limitations:
The culturally specific dynamics of gender roles in Peru are not deeply interrogated, which may affect cross-cultural applicability. AGB is theorized entirely from women’s reports. Including male perspectives could enhance understanding of intent and perception.
Recommendations:
Accept with Minor Revisions
The manuscript offers a timely and innovative contribution to behavioral sciences and gender studies. Its methodological robustness, theoretical advancement, and applied relevance make it highly suitable for publication. The manuscript provides a significant theoretical and empirical contribution to gender studies and violence research by introducing and validating the Aggressive Gender Backlash (AGB) construct. The English is largely strong, with clear structure and effective communication of complex ideas. However, the paper would benefit from minor language revisions to improve conciseness, clarity, consistency, and academic tone.
Minor revisions should focus on:
- Elaborating on the cultural and contextual constraints of the study.
- Suggesting avenues for cross-cultural validation.
- Clarifying terminology to prevent misinterpretation of "aggressive" in non-violent contexts.
- Before publication, a light professional language edit is recommended to ensure idiomatic fluency and polish.
The manuscript is well-written overall, with clear academic structure and precise terminology. However, there are several areas where the English could be refined to enhance clarity, flow, and academic tone. Below are some observations and suggestions:
Consistency
Original:
“Justifica la creación de una escala específica.”
Suggestion:
Translate to English consistently:
“This justifies the development of a specific measurement scale.”
Rationale: Maintain language consistency throughout.
Verb Tense Consistency
For example:
“We postulate that the Aggressive Gender Backlash (AGB) has a semi-circumplex structure.”
Suggestion:
“The study postulates that AGB exhibits a semi-circumplex structure.”
Rationale: Maintain third-person academic tone and verb consistency.
Author Response
Response to Reviewer 3
Dear Reviewer,
We sincerely appreciate your thoughtful, generous, and constructive review of our manuscript titled “The Aggressive Gender Backlash in Intimate Partner Relationships: A Theoretical Framework and Initial Measurement.” Your recognition of the manuscript’s theoretical innovation, methodological rigor, and policy relevance deeply encourages us. We also thank you for the clear and actionable suggestions to enhance the clarity, consistency, and cross-cultural applicability of our work.
Below, we address each of your comments in detail:
- Cultural and contextual constraints
Comment: “The culturally specific dynamics of gender roles in Peru are not deeply interrogated, which may affect cross-cultural applicability.”
Response: We fully agree with this observation. In the revised version, we have elaborated more clearly in both the Limitations and Discussion sections on how the Peruvian context—particularly the microfinance and community banking environment—may influence manifestations and perceptions of AGB. We have also acknowledged how factors such as collectivism, machismo, and community surveillance may shape both the prevalence and tolerance of such behaviors. This contextualization helps readers understand the scope and boundary conditions of our findings.
- Male perspectives and theoretical construction
Comment: “AGB is theorized entirely from women’s reports. Including male perspectives could enhance understanding of intent and perception.”
Response: Thank you for this important point. While our current study is grounded in women's experiences to ensure safety and reliability, we have explicitly acknowledged in the Future Research section the need to include men’s perspectives through qualitative or mixed-method designs. We suggest that applying the General Aggression Model (GAM) to male narratives could enrich the theoretical foundation of AGB and reveal the subjective justifications and meanings attributed to their behaviors.
- Clarification of the term "aggressive"
Comment: “Clarifying terminology to prevent misinterpretation of ‘aggressive’ in non-violent contexts.”
Response: We understand the potential for confusion, particularly in non-specialist or policy audiences. To address this, we now clarify early in the introduction that “aggressive” refers to covert, intentional behaviors that undermine or control another’s autonomy, consistent with psychological aggression literature, and distinct from physical violence. A footnote also addresses this semantic distinction.
- Suggestions for cross-cultural validation
Comment: “Suggesting avenues for cross-cultural validation.”
Response: We have added a paragraph in the Discussion and Future Research sections encouraging comparative studies in diverse sociocultural settings, including highly egalitarian versus traditional societies, to assess measurement invariance and cultural variation in the manifestations of AGB.
- Language consistency (e.g., Spanish sentence in table, verb tenses, terminology)
Response: Thank you for noting these issues. We have:
- Translated the Spanish sentence in Table 12 and reviewed all tables for consistency.
- Revised verb tenses throughout the manuscript for third-person academic consistency (e.g., “The study postulates…” instead of “We postulate…”).
- Improved the academic tone and clarity in all sections by conducting a line-by-line language edit.
- Removed or rephrased potentially confusing terms such as “stereotypy” or “healthily.”
- Recommendation for professional copy-editing
Response: We appreciate this recommendation. The manuscript has undergone an external academic proofreading round focused on fluency, conciseness, and idiomatic accuracy to ensure it meets the highest editorial standards.
We are very grateful for your supportive and insightful comments. They have allowed us to clarify, polish, and strengthen the manuscript substantially. Thank you for your careful reading and encouragement to contribute meaningfully to the literature on gender relations and covert resistance.
Sincerely,
Aristides Vara-Horna
Noelia Rodriguez-Espartal